# Virophages and retrotransposons colonize the genomes of a heterotrophic flagellate

**Thomas Hackl, Sarah Duponchel, Karina Barenhoff, Alexa Weinmann, Matthias G Fischer\***

Max Planck Institute for Medical Research, Department of Biomolecular Mechanisms, Heidelberg, Germany

**Abstract** Virophages can parasitize giant DNA viruses and may provide adaptive anti-giant virus defense in unicellular eukaryotes. Under laboratory conditions, the virophage mavirus integrates into the nuclear genome of the marine flagellate *Cafeteria burkhardae* and reactivates upon super-infection with the giant virus CroV. In natural systems, however, the prevalence and diversity of host-virophage associations has not been systematically explored. Here, we report dozens of integrated virophages in four globally sampled *C. burkhardae* strains that constitute up to 2 % of their host genomes. These <u>e</u>ndogenous <u>ma</u>virus-<u>l</u>ike <u>e</u>lements (EMALEs) separated into eight types based on GC-content, nucleotide similarity, and coding potential and carried diverse promoter motifs implicating interactions with different giant viruses. Between host strains, some EMALE insertion loci were conserved indicating ancient integration events, whereas the majority of insertion sites were unique to a given host strain suggesting that EMALEs are active and mobile. Furthermore, we uncovered a unique association between EMALEs and a group of tyrosine recombinase retrotransposons, revealing yet another layer of parasitism in this nested microbial system. Our findings show that virophages are widespread and dynamic in wild *Cafeteria* populations, supporting their potential role in antiviral defense in protists.

**\*For correspondence:**
mfischer@mr.mpg.de

**Competing interest:** The authors declare that no competing interests exist.

## Introduction

Many eukaryotic genomes harbor endogenous viral elements (EVEs) (*Feschotte and Gilbert, 2012*). For retroviruses, integration as a provirus is an essential part of their replication cycles, but other viruses also occasionally endogenize, for instance with the help of cellular retroelements (*Holmes, 2011*). Some green algal genomes even contain giant EVEs of several hundred kilobase pairs (kbp) in length (*Moniruzzaman et al., 2020*), but unlike prophages in bacteria and archaea, most eukaryotic EVEs are thought to be 'genomic fossils' and incapable of virion formation and horizontal transmission. However, some viral genes may be co-opted for various host functions (*Frank and Feschotte, 2017*; *Aswad and Katzourakis, 2012*). In recent years, the exploration of protist-infecting giant viruses has uncovered a novel class of associated smaller DNA viruses with diverse and unprecedented genome integration capabilities.

Viruses of the family *Lavidaviridae*, commonly known as virophages, depend for their replication on giant DNA viruses of the family *Mimiviridae* and can parasitize them during coinfection of a suitable protist host (*La Scola et al., 2008*; *Krupovic et al., 2016a*; *Duponchel and Fischer, 2019*). A striking example is the virophage mavirus, which strongly inhibits virion synthesis of the lytic giant virus CroV during coinfection of the marine heterotrophic nanoflagellate *Cafeteria* sp. (Stramenopiles; Bicosoe-cida) (*Fischer and Suttle, 2011*; *Fischer and Hackl, 2016*). Virophages possess 15–30 kbp long double-stranded (ds) DNA genomes of circular or linear topology that tend to have low GC-contents

**eLife digest** Viruses exist in all ecosystems in vast numbers and infect many organisms. Some of them are harmful but others can protect the organisms they infect. For example, a group of viruses called virophages protect microscopic sea creatures called plankton from deadly infections by so-called giant viruses. In fact, virophages need plankton infected with giant viruses to survive because they use enzymes from the giant viruses to turn on their own genes.

A virophage called mavirus integrates its genes into the DNA of a type of plankton called *Cafeteria.* It lays dormant in the DNA until a giant virus called CroV infects the plankton. This suggests that the mavirus may be a built-in defense against CroV infections and laboratory studies seem to confirm this. But whether wild *Cafeteria* also use these defenses is unknown.

Hackl et al. show that virophages are common in the DNA of wild *Cafeteria* and that the two appear to have a mutually beneficial relationship. In the experiments, the researchers sequenced the genomes of four *Cafeteria* populations from the Atlantic and Pacific Oceans and looked for virophages in their DNA. Each of the four *Cafeteria* genomes contained dozens of virophages, which suggests that virophages are important to these plankton. This included several relatives of the mavirus and seven new virophages. Virophage genes were often interrupted by so called jumping genes, which may take advantage of the virophages the way the virophages use giant viruses to meet their own needs.

The experiments show that virophages often co-exist with marine plankton from around the world and these relationships are likely beneficial. In fact, the experiments suggest that the virophages may have played an important role in the evolution of these plankton. Further studies may help scientists learn more about virus ecology and how viruses have shaped the evolution of other creatures.

(27–51%) (*Fischer, 2020*). A typical virophage genome encodes 20–30 proteins, including a major capsid protein (MCP), a minor capsid or penton protein (PEN), a DNA packaging ATPase, and a maturation cysteine protease (PRO) (*Krupovic et al., 2016a*). In addition to this conserved morphogenesis module, virophages encode DNA replication and integration proteins that were likely acquired independently in different virophage lineages (*Yutin et al., 2013*). Viruses in the genus *Mavirus* contain a *rve*-family integrase (rve-INT) that is also found in retrotransposons and retroviruses, with close homologs among the eukaryotic Maverick/Polinton elements (MPEs) (*Fischer and Suttle, 2011*). MPEs were initially described as DNA transposons (*Pritham et al., 2007*; *Kapitonov and Jurka, 2006*), but many of them carry the morphogenesis gene module and thus qualify as endogenous viruses (*Krupovic et al., 2014*).

Phylogenetic analysis suggests that mavirus-type virophages share a common ancestry with MPEs and the related Polinton-like viruses (PLVs) (*Fischer and Suttle, 2011*; *Yutin et al., 2013*). We therefore tested the integration capacity of mavirus using the cultured protist *Cafeteria burkhardae* (formerly *Cafeteria roenbergensis*; *Fenchel and Patterson, 1988*; *Schoenle et al., 2020*) and found that mavirus integrates efficiently into the nuclear host genome (*Fischer and Hackl, 2016*). The resulting mavirus provirophages are transcriptionally silent unless the host cell is infected with CroV, which leads to reactivation and virion formation of mavirus. Newly produced virophage particles then inhibit CroV replication and increase host population survival during subsequent rounds of coinfection (*Fischer and Hackl, 2016*). The mutualistic *Cafeteria*-mavirus symbiosis may thus act as an adaptive defense system against lytic giant viruses (*Fischer and Hackl, 2016*; *Koonin and Krupovic, 2016*). The integrated state of mavirus is pivotal to the proposed defense scheme as it represents the host's indirect antigenic memory of CroV (*Koonin and Krupovic, 2017*). We therefore investigated endogenous virophages to assess the prevalence and potential significance of virophage-mediated defense systems in natural protist populations.

Here, we report that mavirus-like EVEs are common, diverse, and most likely active mobile genetic elements (MGEs) of *C. burkhardae*. Our results suggest an influential role of these viruses on the ecology and evolution of their bicosoecid hosts.

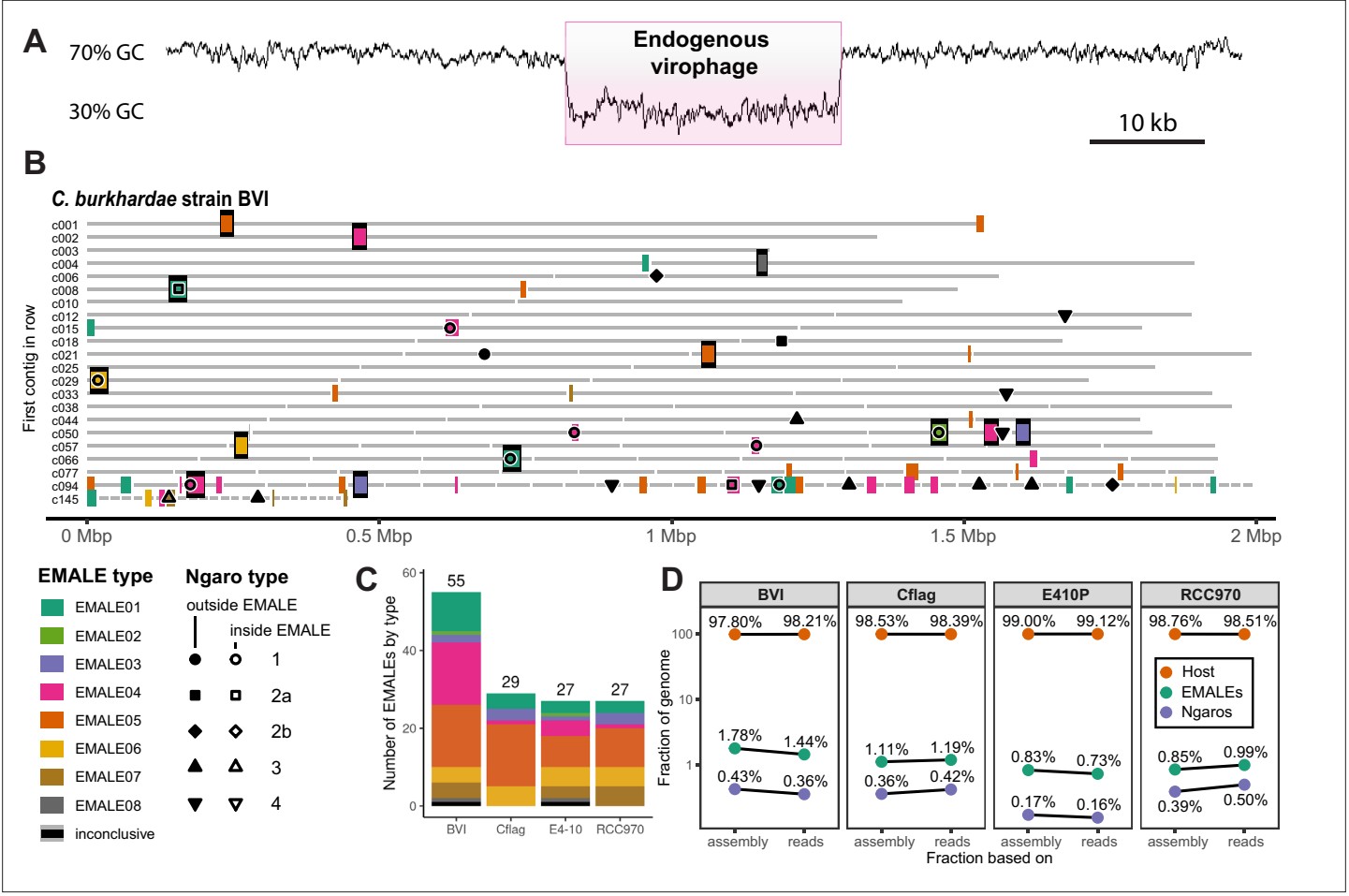

**Figure 1.** Endogenous virophages in *Cafeteria burkhardae*. (**A**) GC-content graph signature of a virophage element embedded in a high-GC host genome. Shown is a region of contig BVI_c002 featuring an integrated virophage (pink box) flanked by host sequences. (**B**) Location of partial or complete virophage genomes and Ngaro retrotransposons in the genome assemblies of *C. burkhardae* strain BVI (see *Figure 1—figure supplement 1* for all four strains). Horizontal lines represent contigs of decreasing length ordered from left to right and from top to bottom, with numbers shown for the first contig of each line; colored boxes indicate endogenous mavirus-like elements (EMALEs). Fully assembled elements are framed in black. Ngaro retrotransposon positions are marked by black symbols; open symbols indicate Ngaros integrated inside a virophage element. (**C**) Graphic summary of the number and types of all EMALEs identified in each of the four *C. burkhardae* strains. (**D**) Nucleotide contributions of EMALEs and Ngaros to *Cafeteria* genomes. Fractions for each strain are computed based on nucleotides in the assembly (left) and nucleotides in the reads (right) mapping to the different parts of the assembly.

The online version of this article includes the following figure supplement(s) for figure 1:

**Figure supplement 1.** Distribution of EMALE and Ngaro retrotransposon integration sites in the four *Cafeteria burkhardae* genome assemblies.

## Results

### Endogenous virophages are abundant in *Cafeteria* genomes

In preparation of screening for endogenous virophages, we generated high-quality de novo genome assemblies of four cultured *C. burkhardae* strains (*Hackl et al., 2020*). These strains, designated BVI, Cflag, E4-10P (E4-10), and RCC970-E3 (RCC970), were isolated from the Caribbean Sea in 2012, the Northwest Atlantic in 1986, the Northeast Pacific in 1989, and the Southeast Pacific in 2004, respectively. We sequenced their genomes using both short-read (Illumina MiSeq) and long-read (Pacific Biosciences RSII) technologies in order to produce assemblies that would resolve 20–30 kb long repetitive elements within the host genomic context. Each *C. burkhardae* genome assembly comprised of 34–36 megabase pairs with an average GC-content of 70 % (*Hackl et al., 2020*).

To identify endogenous virophages, we combined sequence similarity searches against known virophage genomes with genomic screening for GC-content anomalies. The two approaches yielded

redundant results and virophage elements were clearly discernible from eukaryotic genome regions based on their low (30–50%) GC-content (*Figure 1A*). Each element had at least one open reading frame (ORF) with a top blastp hit to a mavirus protein, with no elements bearing close resemblance to Sputnik or other virophages outside the genus *Mavirus*. In the four *Cafeteria* genomes combined, we found 138 endogenous mavirus-like elements (EMALEs, *Figure 1B and C*; *Figure 1—figure supplement 1*, *Supplementary file 1*). Thirty-three of these elements were flanked by terminal inverted repeats (TIRs) and host DNA and can thus be considered full-length viral genomes.

The remainder were partial virophage genomes that were located at contig ends or on short contigs. These cases arise from incomplete assembly rather than from biological truncations, since the assembly algorithm probably terminated due to the presence of multiple identical or highly similar EMALEs within the same host genome – a well-known issue for repetitive sequences (*Kolmogorov et al., 2019*). With 55 elements, *C. burkhardae* strain BVI contained nearly twice as many EMALEs as any of the other strains, where we found 27–29 elements per genome (*Figure 1C*, *Supplementary file 1*). Compared to the total assembly length, EMALEs composed an estimated 0.7–1.8% of each host assembly (*Figure 1D*). Contributions calculated from assemblies deviated only slightly (0.01–0.3%) from read-based calculations. Therefore, the assemblies seem to provide a good representation of the actual contribution of EMALEs to the overall host genomes.

## EMALEs are genetically diverse

From here on, we focus our analysis on the 33 complete EMALE genomes, which were 5.5–21.5 kb long with a median length of 19.8 kb, and TIRs that varied in length from 0.2 to 2.3 kb with a median of 0.9 kb (*Supplementary file 1*, *Figure 3—figure supplement 1*). Their GC-contents ranged from 29.7% to 52.7%, excluding retrotransposon insertions where present. To classify EMALEs we used an all-versus-all DNA dot plot approach (*Figure 2*). It revealed two main blocks: The first block contained EMALEs with GC-contents of 29.7–38.5% (median 35.3%), whereas EMALEs in the second block had GC-contents ranging from 47.2% to 52.7% (median 49.3%). The *C. burkhardae* EMALEs can thus be roughly separated into low-GC and mid-GC groups.

Based on the similarity patterns within each block, we further distinguish eight EMALE types, with low-GC EMALEs comprising types 1–4 and mid-GC EMALEs comprising types 5–8 (*Figure 2*). Representative genome diagrams for each EMALE type are shown in *Figure 3*, for a schematic of all 33 complete EMALEs, see *Figure 3—figure supplement 1*. According to this classification scheme, the reference mavirus strain Spezl falls within type 4 of the low-GC EMALEs (*Figures 2 and 3*, *Figure 3—figure supplement 1*). Partial EMALEs were classified based on their sequence similarity to full-length type species (*Figure 2—figure supplement 1*).

The codon and amino acid composition of EMALE genes clearly correlated with the overall GC-content of the EMALE genomes (*Figure 2—figure supplement 2*). For each encoded amino acid, we observed a strong shift toward synonymous codons reflecting the overall GC trend, and across amino acids, we observed a shift from those encoded by high-GC codons to those encoded by low-GC codons in low-GC EMALEs and vice versa. This uniform trend across all amino acids likely indicates that selection and evolutionary processes driving GC-content variation in these viruses act on the nucleotide level, rather than on the encoded proteins.

With few exceptions, EMALEs are predicted to encode 17–21 proteins each. None of the encoding genes was found to contain introns. The virion morphogenesis module in EMALE types 1 and 3–7 consists of the canonical virophage core genes corresponding to MCP, PEN, ATPase, and PRO proteins. Type 2 EMALEs likely encode a different set of capsid genes as discussed below, and the truncated EMALE type 8 lacks recognizable morphogenesis genes. Another highly conserved gene in EMALE types 1 and 3–7 is *MV14*, which is always found immediately upstream of the *ATPase* (*Figure 3*, *Figure 3—figure supplement 1*) and codes for a protein of unknown function that is part of the mavirus virion (*Born et al., 2018*). *MV14* is present in various metagenomic virophage sequences (*Paez-Espino et al., 2019*) and, based on synteny and protein localization, likely encodes an important virion component in members of the genus *Mavirus*.

The replication/integration module consists of the *rve-INT* gene and at least one additional ORF coding for a primase/helicase and a DNA polymerase. Low-GC EMALEs encode a mavirus-related primase/helicase and protein-primed family B DNA polymerase (pPolB) (*Figure 3*, *Figure 3—figure supplement 1*). Mid-GC EMALEs, on the other hand, lack the *pPolB* gene and feature a longer

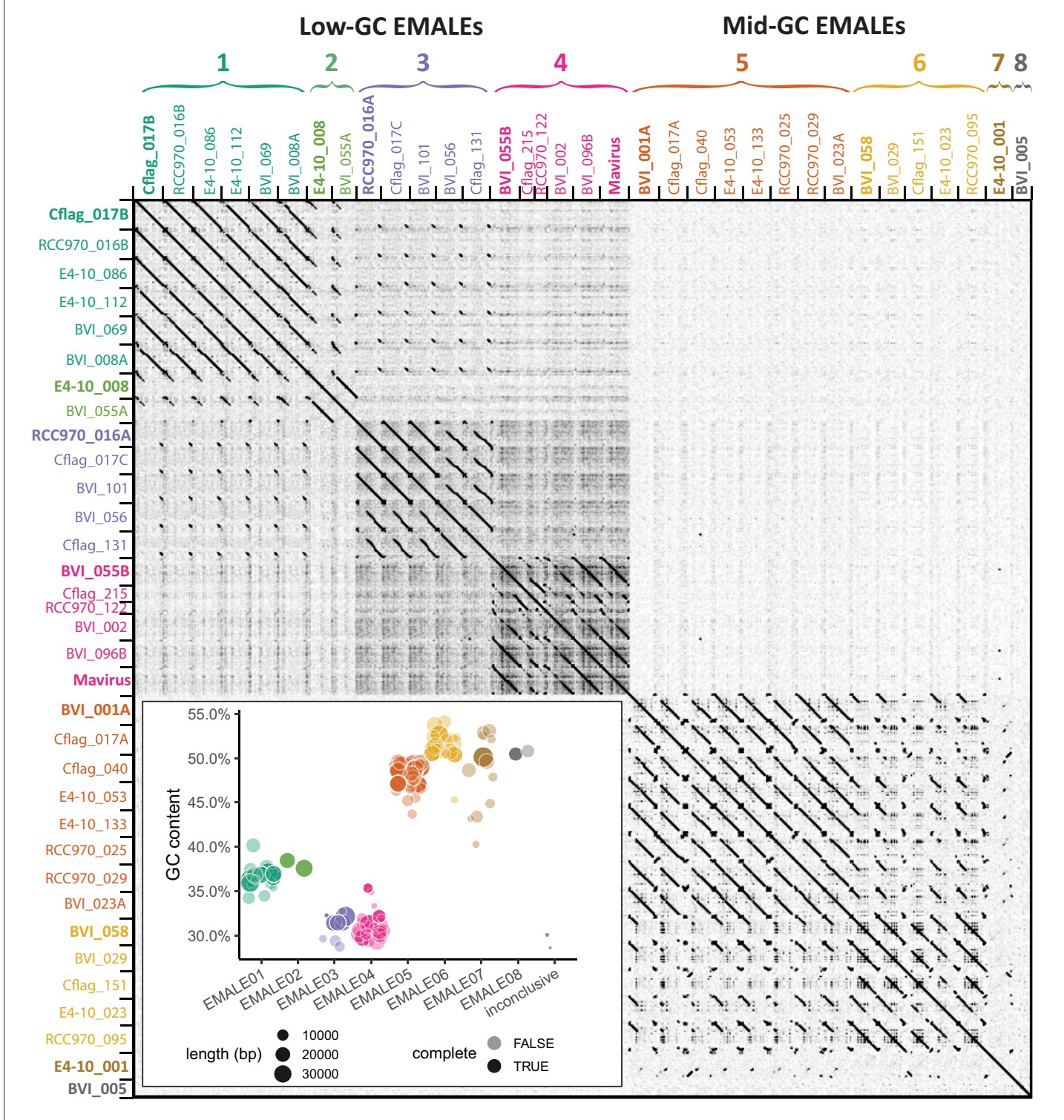

**Figure 2.** Classification of endogenous virophages based on DNA dot plot analysis. The self-versus-self DNA dot plot of concatenated sequences of 33 complete EMALE genomes and mavirus reveals two main block patterns, corresponding to EMALEs with low (29–38%) GC-content and medium (47–53%) GC-content. Smaller block patterns define EMALE types 1–8. EMALE identifiers indicate the host strain and contig number where the respective element is found. Multiple EMALEs on a single contig are distinguished by terminal letters. Elements printed in bold represent the type species shown in *Figure 3*. Inset: GC-content distribution of complete and partial EMALEs labeled 'complete: TRUE/FALSE'. Some partial EMALEs were too short for type assignment and are thus inconclusive. Retrotransposon insertions, where present, were removed prior to analysis.

The online version of this article includes the following figure supplement(s) for figure 2:

*Figure 2 continued on next page*

*Figure 2 continued*

**Figure supplement 1.** Type assignment for incomplete EMALEs.

**Figure supplement 2.** Codon usage with respect to GC-content in different EMALE types.

primase/helicase ORF that may include a DNA polymerase domain similar to the helicase-polymerase fusion genes described in PLVs (*Krupovic et al., 2016b*).

Other mavirus genes frequently found in EMALEs include *MV19* (encoding a putative protease domain), and two genes of unknown function, *MV08* and *MV12*. Interestingly, all mid-GC EMALEs encode a predicted tyrosine recombinase (YR) in addition to the *rve-INT* and thus possess two predicted enzymes for genome integration. YRs have been found in other virophages and likely catalyze integration into giant virus genomes (*Desnues et al., 2012*; *Yutin et al., 2015*). Notable genes unique to one EMALE type include a putative DNA methylase and a ribonucleotide reductase small subunit gene found in EMALE07. The Tlr6F protein encoded by EMALE types 1 + 2 is present in diverse MGEs, including other virophages, PLVs, and large DNA viruses of the phylum *Nucleocytoviricota* (*Koonin and Krupovic, 2017*; *Stough et al., 2019*).

In general, genes were syntenic between EMALEs of the same type, whereas gene order was poorly conserved among EMALEs of different types, with the following exceptions: *MCP* was always preceded by *PEN*, and *ATPase* was always preceded by *MV14*, whereas the *MV14-ATPase-PRO-PEN-MCP* morphogenesis gene order as seen in mavirus was present only in EMALE types 4–7. EMALE02 represents an interesting case, as it shares 6–7 kb of its 5' part (we chose the *primase/helicase* genes to mark the 5' end of all EMALEs) with EMALE01, while the remaining 11 kb are not closely related to other EMALEs or virophages (*Figure 3—figure supplement 2*). Genes encoded in the latter region are mostly ORFans, with the exception of an *MV12*-like gene and divergent *MCP* and *ATPase* genes with remote similarity to PLVs (*Bellas and Sommaruga, 2021*) and adintoviruses (*Starrett et al., 2021*). EMALE02 may thus be the result of a recombination event that exchanged the canonical virophage morphogenesis module of EMALE01 with capsid genes of a PLV (*Figure 3*, dashed line). Overall, these observations support the notion that recombination and non-homologous gene replacement are important factors in virophage genome evolution (*Yutin et al., 2013*).

## Core gene conservation and non-homologous gene replacement in EMALEs

To validate our classification scheme for EMALEs and to place them in a phylogenetic context to other virophages, we used maximum likelihood reconstruction on the core proteins MCP, PEN, ATPase, and PRO, as well as on rve-INT (*Figure 4*). In the resulting phylogenetic trees, EMALE core proteins formed monophyletic clades with mavirus and related sequences from environmental samples, thus significantly expanding the known diversity of the genus *Mavirus*. The environmental sequences that clustered with EMALE core proteins include a single amplified genome (SAG) from an uncultured chrysophyte (*Castillo et al., 2019*), the metagenomic Ace Lake Mavirus (ALM) (*Zhou et al., 2013*), and four additional metagenomes that were identified in a global survey of virophage sequences (*Paez-Espino et al., 2019*). The chrysophyte SAG is nearly identical to mavirus strain Spezl and indicates that the host range of mavirus extends beyond bicosoecids. The metagenomic sequences either clustered with one of the EMALE types, or branched separately from them, which suggests the existence of additional sub-groups (e.g. M590M2_1006461).

Within the *Mavirus* clade, EMALEs of a given type were monophyletic for each of the four core proteins, which corroborates their dot plot-based classification. It is worth noting that although EMALEs of types 5 and 6 are largely syntenic (*Figure 3*, *Figure 3—figure supplement 1*), they were clearly distinguishable in their phylogenetic signatures (*Figure 4*). A comparison of clade topologies revealed that even within the conserved morphogenesis module, individual proteins differed with regard to their neighboring clades, and low-GC and mid-GC EMALEs did not cluster separately from each other. These observations could suggest that the morphogenesis modules of different EMALE types diversified simultaneously and that adaptation of GC-content may occur rather quickly.

In contrast, phylogenetic analysis of rve-INT proteins revealed separate clades for low-GC and mid-GC EMALEs (*Figure 4*). Each of these clades was affiliated with different cellular homologs that included MPEs and retroelements. Notably, the *rve-INT* genes of low-GC EMALEs were located near

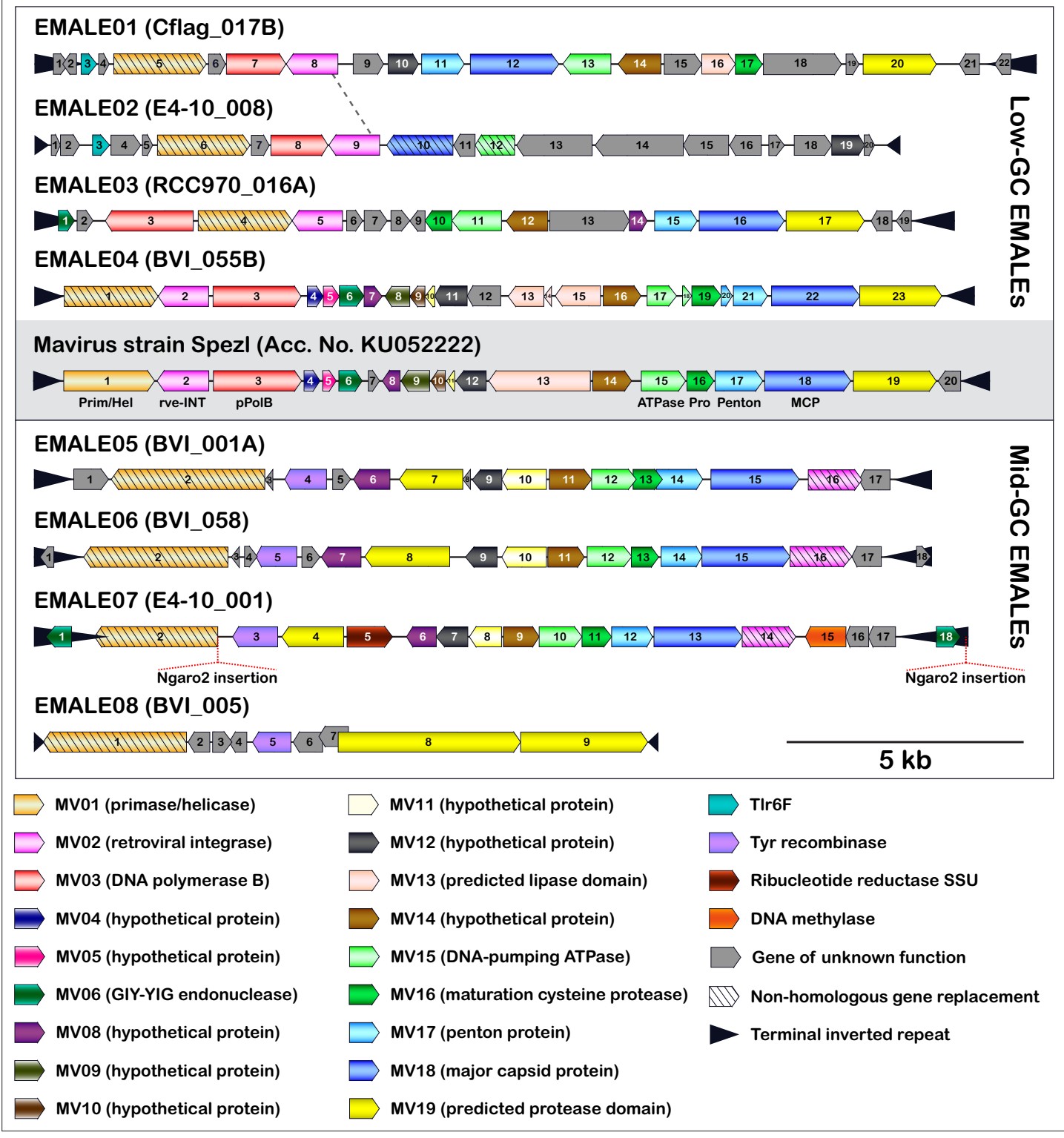

**Figure 3.** Genome organization of eight EMALE types found in *Cafeteria burkhardae*. Shown are schematic genome diagrams of the EMALE type species 1–8; for all 33 complete EMALEs, see *Figure 3—figure supplement 1*. The reference mavirus genome with genes *MV01-MV20* is included for comparison. Homologous genes are colored identically; genes sharing functional predictions but lacking sequence similarity to the mavirus homolog are hatched. Open reading frames are numbered individually for each element. Ngaro retrotransposon insertion sites are indicated where present. The dotted line between EMALE01 and EMALE02 separates a homologous region (left) from unrelated DNA sequences (right) and thus indicates the location of a probable recombination event.

*Figure 3 continued on next page*

*Figure 3 continued*

The online version of this article includes the following figure supplement(s) for figure 3:

**Figure supplement 1.** Coding capacity of 33 completely assembled EMALEs in *Cafeteria burkhardae*.

**Figure supplement 2.** Partial synteny between EMALE01 and EMALE02.

**Figure supplement 3.** Unique and orthologous EMALE integration loci among four *Cafeteria* strains.

**Figure supplement 4.** DNA dot plots of selected EMALE loci as shown in *Figure 3—figure supplement 3*.

**Figure supplement 5.** Putative promoter motifs in EMALE genomes.

**Figure supplement 6.** Correction of Illumina/PacBio-based assemblies by PCR and Sanger sequencing.

the 5' end of the genomes, whereas in mid-GC EMALEs, they were located near the 3' end (*Figure 3*, *Figure 3—figure supplement 1*). These observations suggest that EMALEs encode two different rve-INT versions, one specific for low-GC EMALEs that co-occurs with the pPolB and a shorter primase/helicase ORF, and one specific for mid-GC EMALEs that co-occurs with a longer primase/helicase ORF. The two integrase versions may have been acquired independently, or one version could have replaced the other during EMALE evolution. Such non-homologous gene replacement appears to have taken place among the primase/helicase genes, too, as previously noted for virophages in general (*Yutin et al., 2013*). EMALEs encode several different versions of primase/helicase genes with a degree of

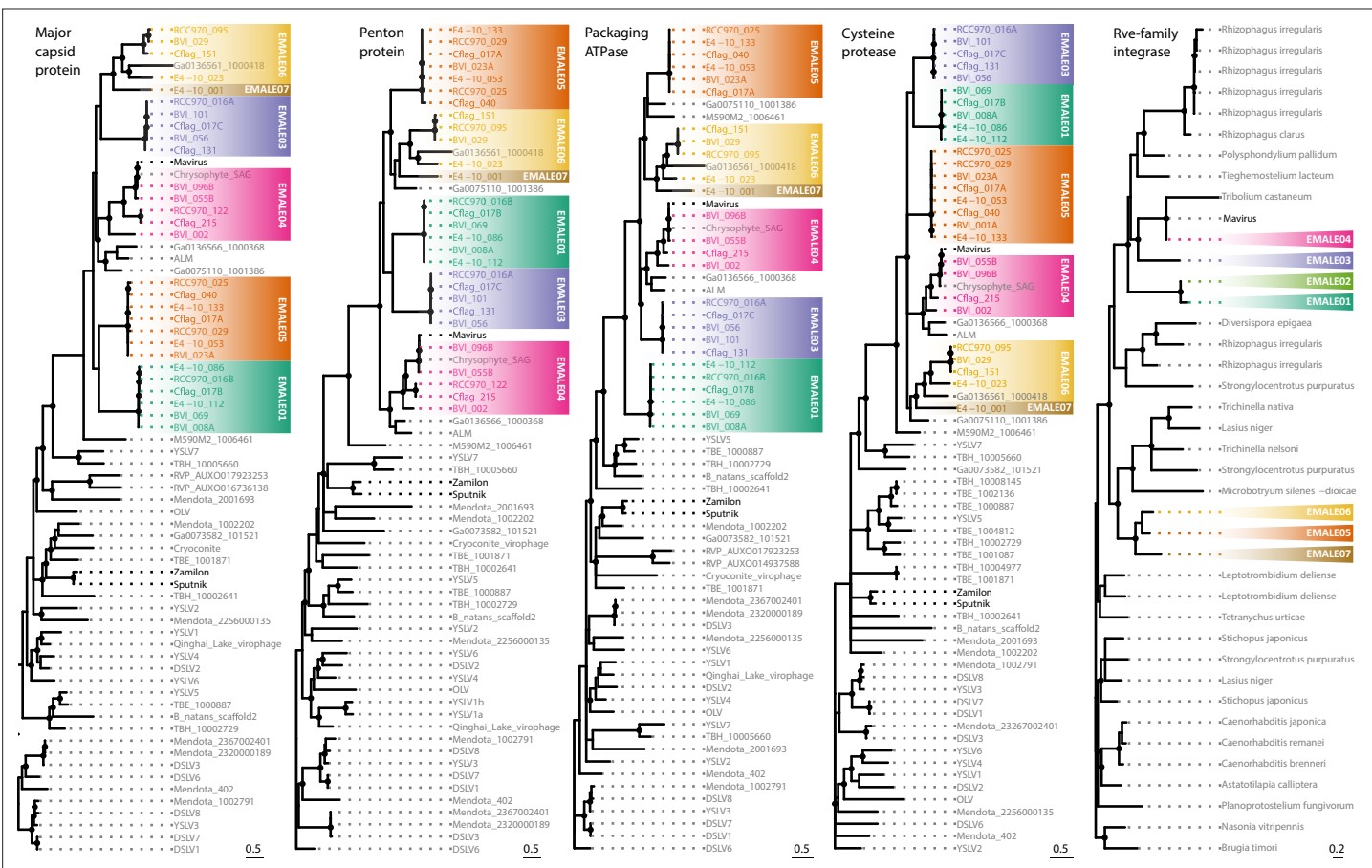

**Figure 4.** Phylogenetic reconstruction of conserved EMALE proteins. Unrooted maximum likelihood trees were constructed from multiple sequence alignments of the four virophage core proteins major capsid protein (MCP), penton protein (PEN), ATPase, and protease (PRO), as well as of the retroviral integrase. Nodes with bootstrap values of 80 % or higher are marked with dots. EMALEs are color-coded by type; cultured virophages are printed in bold. ALM, Ace Lake Mavirus; DSLV, Dishui Lake virophage; OLV, Organic Lake virophage; RVP, rumen virophage; TBE/TBH, Trout Bog Lake epi-/hypolimnion; YSLV, Yellowstone Lake virophage. Metagenomic sequences starting with Ga and M590 are derived from *Paez-Espino et al., 2019*.

The online version of this article includes the following figure supplement(s) for figure 4:

**Figure supplement 1.** Maximum likelihood reconstruction of EMALE tyrosine recombinase phylogenies.

amino acid divergence that precluded their inclusion in a single multiple sequence alignment. The YR proteins encoded by EMALE types 5–8 formed a monophyletic clade and were part of a larger group of recombinases that included virophages from freshwater metagenomes, as well as microalgae and algal nucleocytoviruses (*Figure 4—figure supplement 1*).

## *Cafeteria* strains differ in their EMALE composition

The four *C. burkhardae* strains displayed distinct EMALE signatures: strain BVI had the highest number of virophage elements with 13 complete and 42 partial EMALEs, whereas the other three strains had 6–7 complete and 20–22 partial EMALEs each (*Figure 1C*, *Supplementary file 1*). EMALE types 1, 3, 4, 5, and 6 were present in every host strain, EMALE07 was found in all strains except Cflag, and EMALE types 2 and 8 were detected in strains BVI and E4-10 only. We found no evidence for sequence-specific genome integration of EMALEs after inspecting the host DNA sequences that flanked EMALE integration sites, which confirms previous reports of mavirus integration (*Fischer and Hackl, 2016*). EMALEs were flanked by target site duplications (TSDs) that were predominantly 3–5 bp in length, although some were as short as 1 bp or as long as 9 bp (*Supplementary file 1*). By comparison, mavirus and MPEs generate 5–6 bp long TSDs upon integration (*Fischer and Hackl, 2016*; *Pritham et al., 2007*; *Kapitonov and Jurka, 2006*).

To assess whether homologous EMALEs were found in identical loci in closely related host genomes, we conducted sequence similarity searches with the flanking regions of each of the 33 fully resolved EMALEs. Whenever these searches returned a homologous full or partial EMALE with at least one matching host flank, we considered the EMALE locus to be conserved in these host strains. We found varying degrees of conservation, with examples shown in *Figure 3—figure supplement 3*. In 11 cases, an EMALE insertion was conserved in at least two host strains (*Supplementary file 1*): three EMALE loci were shared by all four strains, four were shared by three strains, and another four were shared by two strains. Based on conserved EMALE loci, strains Cflag and RCC970 were most closely related with nine shared EMALE integrations, which is in line with phylogenetic and average nucleotide identity (ANI) analyses of these strains (*Hackl et al., 2020*). The four *C. burkhardae* genomes have ANIs of >99% and thus appear to differ mostly based on their content of EMALEs and other MGEs.

The most parsimonious scenario for the origin of EMALEs that are located in identical loci in different host strains is that they derived from a single integration event. For instance, EMALE03 BVI_101 is orthologous to Cflag_017 C and RCC970_016 A (*Figure 3—figure supplement 3C*), which suggests that this element initially colonized the common ancestor of *C. burkhardae* strains BVI, Cflag, and RCC970. Further cases of redundant EMALEs are Cflag_017B & RCC970_016B (EMALE01) and BVI_029 & RCC970_095 (EMALE06). These elements may thus derive from relatively ancient integration events, whereas 18 of the 33 complete EMALEs represent integrations that were unique to a single host strain (*Supplementary file 1*). Strain BVI contained 10 of these 18 unique integrations, more than twice as many as any other strain.

The genomic landscape around EMALE integration sites ranged from repeat-free flanking regions to complex host repeats (*Figure 3—figure supplement 4*). Of the 29 different integration sites represented by the 33 fully resolved EMALEs, 18 were located near repetitive host DNA (within 10 kb from the insertion site). These repeats, in addition to EMALE TIRs, multiple copies of the same EMALE type, and the putative heterozygosity of EMALE insertions, occasionally caused assembly problems, as illustrated in *Figure 3—figure supplement 3*.

Next, we analyzed whether EMALE insertions interrupted coding sequences of the host. Fifteen integration sites were located within a predicted host gene (13 in exons, 2 in introns), four were found in predicted 3' untranslated regions, and three were located in intergenic regions (*Supplementary file 1*). These data show that EMALE insertions may disrupt eukaryotic genes with potential negative consequences for the host. The apparent preference for integration in coding regions could be assembly related, driven by increased accessibility of euchromatin, or linked to host factors that could direct the rve-INT via its CHROMO domain (*Gao et al., 2008*).

## EMALEs are predicted to be functional and mobile

Based on genomic features such as coding potential, ORF integrity, and host distribution, most EMALEs appear to be active MGEs. With the exception of EMALE08 and EMALE02, all endogenous *Cafeteria* virophages encode the canonical morphogenesis gene module consisting of *MCP*, *PEN*,

*ATPase*, *PRO*, as well as *MV14*. EMALE02 likely encodes more distantly related capsid genes. Therefore, all EMALE types except EMALE08 should be autonomous for virion formation. In addition, all EMALEs contain at least one predicted enzyme for genome integration, an rve-INT in EMALE types 1–7 and a YR in EMALE types 5–8. EMALEs thus encode the enzymatic repertoire for colonizing new host genomes. Finally, the high variability of EMALE integration loci among otherwise closely related host strains strongly argues for ongoing colonization of natural *Cafeteria* populations by virophages.

The genomic similarity to mavirus implies that EMALEs may also depend on a giant virus for activation and horizontal transmission. Shared regulatory sequences in virophages and their respective giant viruses suggest that the molecular basis of virophage activation lies in the recognition of virophage gene promoters by giant virus encoded transcription factors (*Fischer and Suttle, 2011*; *Claverie and Abergel, 2009*; *Legendre et al., 2010*). We therefore analyzed the 100 nt upstream regions of EMALE ORFs for conserved sequence motifs using MEME (*Bailey et al., 2009*). For all type 4 EMALEs, which include mavirus, we recovered the previously described mavirus promotor motif 'TCTA', flanked by AT-rich regions. This motif corresponds to the conserved late gene promoter in CroV (*Fischer and Suttle, 2011*; *Fischer et al., 2010*), thus possibly indicating that all type 4 EMALEs could be reactivated by CroV or close relatives. EMALEs of other types lacked the 'TCTA' motif, but contained putative promoter sequences that may be compatible with different giant viruses (*Figure 3—figure supplement 5*).

MGEs are prone to various decay processes including pseudogenization, recombination, and truncation. Among the 33 fully resolved EMALEs are three truncated elements: Cflag_215 and RCC970_122 (both EMALE04), and BVI_005 (EMALE08) (*Figure 3—figure supplement 1*). Interestingly, even these shorter elements are flanked by TIRs, which must have regenerated after the truncation event. Whereas most EMALE ORFs appeared to be intact, as judged by comparison with homologous genes on syntenic elements, several EMALEs contained fragmented ORFs (e.g. *ATPase* and *PEN* genes in EMALE04 BVI_055B, *Figure 3*). To test whether premature stop codons may be the result of assembly artifacts, we amplified selected EMALEs by PCR and analyzed the products using Sanger sequencing. When we compared the Sanger assemblies with the Illumina/PacBio assemblies, we noticed that the latter contained several substitutions and indels. For example, the MCP gene of EMALE01 RCC970_016B was split into three ORFs in the Illumina/PacBio assembly, whereas a single ORF was present in the corresponding Sanger assembly (*Figure 3—figure supplement 6*). None of the Sanger assemblies confirmed fragmentation of conserved virophage genes, emphasizing the importance of independent sequence validation. However, since it was not possible to re-sequence all potential pseudogene locations, we cannot exclude that some EMALE genes may be fragmented. Overall, EMALE ORFs appeared to be intact and are thus likely to encode functional proteins.

## Tyrosine recombinase retrotransposons integrate into EMALEs

Strikingly, about one in four virophage elements was interrupted by a GC-rich sequence with a typical length of ~6 kb (*Figure 5A*, *Supplementary file 1*). We identified these insertions as retrotransposons of the Ngaro superfamily, within the DIRS order of retrotransposons (*Wicker et al., 2007*). Ngaro retrotransposons feature split direct repeats with $A_1$–[ORFs]–$B_1A_2B_2$ structure (*Figure 5D*), encode a YR instead of the rve-INT that is typical for retrotransposons, and are found in various eukaryotes from protists to vertebrates (*Poulter and Goodwin, 2005*). In the four *C. burkhardae* strains, we annotated 80 Ngaro elements, with 10–25 copies per strain (*Supplementary file 1*). In addition, we found isolated AB repeats scattered throughout the genome, which could have arisen from recombination of 5' and 3' repeats and are reminiscent of solo long terminal repeats of endogenous retroviruses (*Friedli and Trono, 2015*). Solo AB repeats were also occasionally present in EMALEs (BVI_016, BVI_115, Cflag_024, Cflag_214, Cflag_215). Dot plot analysis of concatenated *Cafeteria* Ngaro sequences revealed four distinct types that showed no similarity at the nucleotide level, but appeared to share the same coding potential (*Figure 5B and D*). Based on synteny to previously described Ngaros, ORF1 may encode a Gag-like protein; ORF2 encodes predicted reverse transcriptase (Pfam PF00078) and ribonuclease H domains (Pfam PF17919); and ORF3 encodes a predicted YR (Pfam PF00589) with the conserved His-X-X-Arg motif and catalytic Tyr residue. Ngaro YRs are related to putative transposons of bacteria and eukaryotes (*Figure 5—figure supplement 1*), but bear no sequence similarity to the EMALE-encoded YRs.

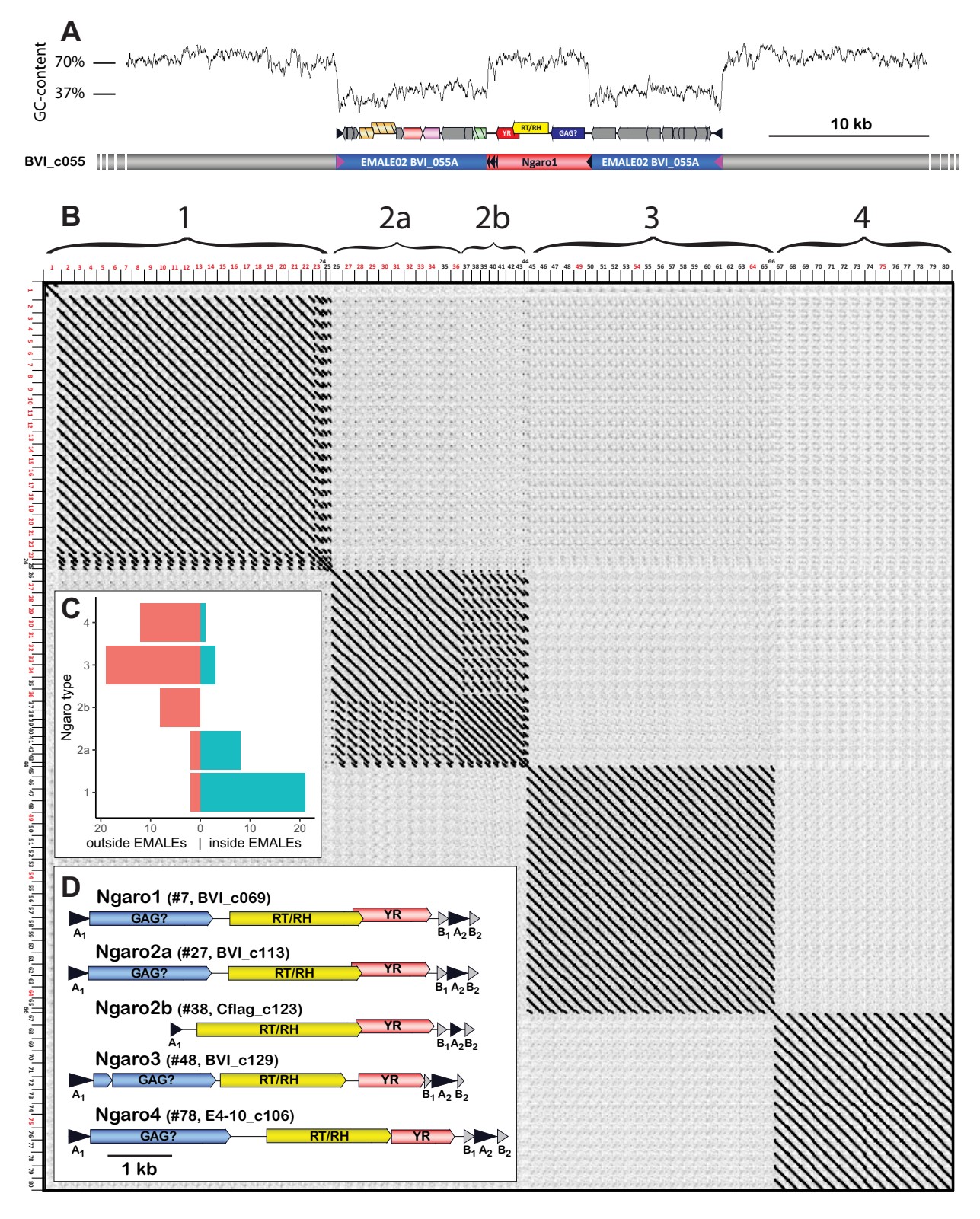

**Figure 5.** Ngaro retrotransposons in *Cafeteria burkhardae*. (**A**) Genomic profile of an EMALE-integrated Ngaro element showing a GC-content graph (top), open reading frame (ORF) organization of EMALE and Ngaro (middle), and a schematic overview of the three genomic entities (bottom; host: gray, EMALE: blue, Ngaro: red). (**B**) Self-versus-self DNA dot plot of 80 concatenated Ngaro sequences. Block patterns define Ngaro types 1–4. Ngaros are numbered according to ***Supplementary file 1***, with red numbers indicating retrotransposons inserted in EMALEs. (**C**) Distribution of Ngaro integration

*Figure 5 continued on next page*

*Figure 5 continued*

loci in EMALE and host DNA. Ngaro types 1 and 2 a show a clear preference for EMALE loci, in contrast to Ngaro types 2b, 3, and 4 that are mostly found in host loci. (**D**) Coding potential of *C. burkhardae* Ngaro retrotransposons, shown for one example per type with their host strain and contig numbers listed. Triangles indicate direct repeats. GAG, group specific antigen; RT, reverse transcriptase; RH, ribonuclease H; YR, tyrosine recombinase.

The online version of this article includes the following figure supplement(s) for figure 5:

**Figure supplement 1.** Phylogenetic placement of Ngaro tyrosine recombinases.

**Figure supplement 2.** Protein length distributions in EMALEs with and without retrotransposons.

**Figure supplement 3.** Nested integration scenario involving one EMALE and three Ngaro retrotransposons.

Interestingly, the four Ngaro types in *C. burkhardae* differed with regard to their integration site preference. Whereas 91 % of type 1 Ngaros were inserted in an EMALE, this was the case for only 14% and 8% of type 3 and type 4 Ngaros, respectively. At first glance, type 2 Ngaros were distributed evenly among viral and eukaryotic DNA (47 % inside EMALEs, 53 % outside EMALEs). However, several type 2 Ngaros were truncated at their 5' end, featuring ~2 kb long deletions that covered ORF1. All of these deletion variants, which we designate as type 2b, were located in host DNA, in contrast to 82 % of the full-length type 2 a Ngaros that were inserted in EMALEs (*Figure 5C and D*). Similarly, most of the type 1 Ngaros that were inserted into eukaryotic chromatin also lacked ORF1. Hence, it appears that ORF1 determines the integration site specificity of Ngaro retrotransposons. We did not observe any preference of host-genome-integrated Ngaros for regions with a GC-content lower than the host average. Out of 20 Ngaro insertion sites with analyzable EMALE flanking regions (including partial EMALEs), 9 were located in intergenic regions, 7 in EMALE genes (1 × *primase/helicase*, 1 × *pPolB*, 1× *MV12*, 4× *MV19*), and 4 in TIRs (*Supplementary file 1*). Considering that intergenic regions comprise only 5–10% of an EMALE genome, we notice a significant bias (p < 1e⁻⁵, Fisher's exact test) toward Ngaro integration in intergenic EMALE DNA, which may be caused either by purifying selection of deleterious Ngaro insertions or by a higher preference for Ngaro integration into EMALE intergenic regions, e.g. due to their lower GC-content.

A possible consequence of retrotransposon insertion is the loss of biological activity and subsequent pseudogenization. However, we found that Ngaro-containing EMALEs did not contain more fragmented genes than Ngaro-free EMALEs (*Figure 5—figure supplement 2*). Whereas the biological properties of Ngaro retrotransposons and their influence on host-virus-virophage dynamics remain to be explored, the EMALE-Ngaro interactions appear to be convoluted. For instance, an EMALE03 genome in strain Cflag is interrupted by two adjacent Ngaro1 insertions, while the EMALE itself is located inside an Ngaro4 element (*Figure 5—figure supplement 3*).

## Discussion

Virophages represent a recently discovered family of eukaryotic dsDNA viruses that possess interesting genome integration properties and have potentially far-reaching eco-evolutionary consequences. Our genomic survey of the marine bicosoecid *C. burkhardae* revealed an unexpected abundance and diversity of endogenous virophages, with dozens of elements in a single host genome. Based on DNA dot plots and phylogenetic analysis, we distinguish eight different types of mavirus-related endogenous virophages. Similar to mavirus, these EMALEs could potentially reactivate and replicate in the presence of a compatible giant virus. Mavirus is proposed to act as an adaptive defense system against CroV in *Cafeteria* populations (*Fischer and Hackl, 2016*; *Koonin and Krupovic, 2016*), and our findings suggest that different types of EMALEs may respond to different giant viruses infecting *Cafeteria*. The assortment of endogenous virophages in a given host genome may thus reflect the giant virus infection history of that population (*Blanc et al., 2015*). Some EMALEs are present in orthologous genomic loci in two or more host strains and likely date back to the common ancestor of these strains. However, at least half of the EMALE insertions are specific to a given host strain and may thus have been acquired relatively recently. Combined with the overall integrity of EMALEs and the conservation of integrase and capsid genes, these findings suggest that endogenous virophages in *C. burkhardae* are active MGEs.

Similar to other MGEs, EMALEs are likely prone to various decay processes including pseudogenization, truncation, and recombination; however, the extent to which these mechanisms affect EMALE

stability is currently unclear. Although nonsense mutations appear to be present in some EMALE genes, re-sequencing of selected genomic regions did not confirm pseudogene loci in those cases (*Figure 3—figure supplement 6*). We found three truncated EMALEs (5–15 kb long), but these are not to be confused with the 105 partial EMALEs that resulted from computational limitations during sequence assembly of these multi-copy elements. Recombination most likely created the EMALE02 variant, with no obvious signs of degradation. Based on our EMALE identification approach (see Materials and methods), we are quite confident that we have not missed more severely degraded EMALEs or more divergent virophages, except perhaps for hypothetical elements lacking any recognizable sequence similarity to annotated virophages and possessing a GC-content similar to that of the host genome. Overall, these observations raise the question of how balance between new virophage integrations and loss of existing EMALEs is achieved to prevent overload of the host genome with mobile DNA. Possible explanations include loss by homologous recombination, for example, between TIRs of the same or similar EMALEs, excision of EMALEs via host- or EMALE-encoded integrases or endonucleases, or mechanisms to limit the uptake of new elements. Given that multiple mavirus genome integrations per host cell can be observed within a period of days (*Fischer and Hackl, 2016*) and that virophage integration rates under natural infection conditions are likely to fall within similar timescales, we hypothesize that the majority of EMALE loss may occur before pseudogenization and degradation can take place. EMALEs would thus have relatively short residence times in the unicellular *Cafeteria* compared to germline EVEs of multicellular eukaryotes (*Barreat and Katzourakis, 2021*).

Horizontal transmission of virophages in natural environments is likely limited by their requirement for two concurrent biological entities, namely a susceptible host cell infected with a permissive giant virus. Similar to temperate bacteriophages, persistence in the proviral state may thus be an essential survival strategy for virophages, as underscored by the abundance of EMALEs in *C. burkhardae*. Endogenous virophages may also be common in eukaryotes outside the order Bicosoecales, although a search in 2015 for provirophages in 1153 eukaryotic genomes found only one clear case in the chlorarachniophyte alga *Bigelowiella natans* (*Blanc et al., 2015*). In contrast to rapidly increasing virophage reports from metagenomic sequence mining in recent years (*Paez-Espino et al., 2019*), we propose that the discovery of host-integrated virophages is still hampered not only by sampling bias, but also by technical limitations. For instance, AT-rich mavirus DNA was severely underrepresented when we sequenced the genome of *C. burkhardae* strain E4-10M1 with the standard Illumina MiSeq protocol (*Fischer and Hackl, 2016*). Additional problems arise during binning and assembly procedures. Although our sequencing and assembly strategy was specifically tailored to endogenous virophages and resolved 33 EMALEs in their host genomic context, dozens of EMALEs were only partially assembled, some may contain assembly errors (*Figure 3—figure supplement 3*), and others may have been missed altogether. Advances in long-read sequencing technologies and assembly algorithms will likely alleviate such problems.

Surprisingly, we found that EMALEs were frequently interrupted by Ngaro retrotransposons, which revealed an additional level of nested parasitism in this microbial system. *Cafeteria* genomes contain four distinct Ngaro types with different affinities for EMALEs. Deletion of ORF1 in type 1 and type 2 Ngaros coincides with a decreased occurrence of these retrotransposons in EMALEs (*Figure 5*). Syntenic ORFs in other Ngaros are predicted to encode a Gag-like structural protein (*Poulter and Butler, 2015*), and Gag proteins of several retroviruses have been linked to integration site specificity (*Lewinski et al., 2006*; *Tobaly-Tapiero et al., 2008*). The putative Gag proteins of *Cafeteria* Ngaros may thus influence whether retrotransposon insertion occurs in an EMALE or in eukaryotic chromatin. So far, retrotransposons have not been described for giant DNA viruses or virophages; however, pandoravirus genomes contain DNA transposons (*Sun et al., 2015*), and a class of 7 kb long DNA MGEs called transpovirons interacts with the particles and genomes of *Acanthamoeba*-infecting mimiviruses and their virophages, apparently without affecting viral replication (*Desnues et al., 2012*; *Jeudy et al., 2020*). It remains to be studied whether Ngaro retrotransposons use reactivated virophages or giant viruses as vehicles for horizontal transmission, and what effect retrotransposon insertion in EMALEs has on the fitness of the virophage, the host cell, and their associated giant viruses.

In conclusion, we show that endogenous virophage genomes are abundant and diverse in the marine heterotrophic protist *C. burkhardae*. These mavirus-like EVEs appear to be active and dynamic MGEs with significant potential to shape the genome evolution of their hosts. We present evidence

for recombination and gene exchange within EMALEs, and a previously unknown affiliation between virophages and YR retrotransposons. Our findings imply an important role for EMALEs in the ecology and evolution of bicosoecids and are in line with the hypothesis that endogenous virophages provide adaptive defense against giant viruses.

## Materials and methods

### *C. burkhardae* cultures and genome sequencing

*C. burkhardae* strains BVI, Cflag, E4-10P, and RCC970-E3 were cultured, and their genomes sequenced and assembled as described previously (*Hackl et al., 2020*).

### Detection and annotation of EMALEs

To identify endogenous virophages, we initially took a manual approach and searched for clearly visible drops in GC-content along contigs when visualized in Artemis Release 16.0.0 (*Rutherford et al., 2000*). Based on our experience we suspected that integrated virophages would have a considerably lower GC-content than the host genomes. For those low-GC regions we then computed DNA dot plots with Gepard (*Krumsiek et al., 2007*) to detect TIRs, predicted ORFs with Artemis, manually curated them into gene calls, and also manually assigned functional annotations based on blastp searches against known virophages and public databases. This resulted in the identification of 33 manually curated, TIR-containing high-confidence ('complete') EMALEs.

Partial EMALEs that could not be fully assembled and were thus located on contig termini were identified based on sequence similarity to annotated virophages and EMALEs using the BLAST suite. Their boundaries were determined based on GC-content or sequence similarity to already annotated TIRs. Partial EMALEs were annotated with prodigal-2.6.3 (*Hyatt et al., 2010*). We pre-trained one gene model for mid-GC and one for high-GC EMALEs, and called genes on those genomes separately. Where present, we masked integrated Ngaro elements with 'N's prior to training and annotation because we observed in trial runs that automated gene prediction could not properly resolve the gene structure of the Ngaros and would produce fragmented and spurious gene calls for these regions. We also explored other gene prediction programs including PHANOTATE (*McNair et al., 2019*) and VGAS (*Zhang et al., 2019*), but found pre-trained prodigal to produce results most consistent with manually curated expert annotations for EMALE genomes.

### Detection of Ngaro retrotransposons

Retrotransposon insertions were first noticed in GC-content graphs as 6–7 kb long GC-rich sequences that interrupted AT-rich EMALEs. DNA dot plot analysis of these regions showed split direct repeats, and conserved domain and tblastx searches predicted RNaseH, reverse transcriptase, and tyrosine recombinase domains. ORFs were annotated in Artemis based on the predicted conserved domains. These initially curated protein sequences and direct repeats were then used to seed blast searches against all four *C. burkhardae* genomes to identify additional Ngaro elements, which were inspected individually and annotated manually.

### Nucleotide contributions of EMALEs and Ngaros to *Cafeteria* genomes

To quantify how much of each host genome is comprised of EMALEs and Ngaros, we applied two complementary strategies: (i) We compared the number of nucleotides annotated as EMALEs and Ngaros in the assemblies to the overall assembly sizes, and (ii) we quantified the number of nucleotides in the PacBio reads that we could assign to either of the three fractions – host, EMALE, and Ngaro. The latter approach is less prone to assembly biases, such as overestimation of contributions for elements only present in one allele, or underestimation of contribution due to collapsed repeated copies, or elements not assembled because of low coverage. We aligned the PacBio reads to the assemblies using minimap2 v2.16 (-x map-pb) (*Li, 2018*) and computed the coverage of the different genomic regions with samtools v1.9 (*Li et al., 2009*). We did not consider Illumina data for the quantification due to an observed sequencing bias against low-GC regions that would lead to lower EMALE estimates.

### Codon usage analysis

To analyze possible correlations between EMALE GC-content and the codon composition of their genes, we counted codons of all genes of complete EMALEs with a custom Perl script and visualized their distribution relative to their GC-content with a custom R script.

## Assignment of EMALE and Ngaro types

We first performed pairwise whole-genome comparisons of all 33 EMALEs plus the reference mavirus genome. Next, we concatenated the EMALE genomes according to their nearest sequence neighbors. We then plotted the resulting concatemer against itself with Gepard (*Krumsiek et al., 2007*) using a word length of 10, and analyzed the similarity patterns. The dot plot-based classification was confirmed by phylogenetic analysis of virophage core genes. A similar approach was used for type assignment of Ngaro retrotransposons.

For partial EMALEs, we assigned types in an automated manner based on the highest cumulative blastx bitscores to typespecies EMALE genomes. EMALEs with cumulative bitscores below 100 were classified as 'inconclusive'. For validation, the results were visualized with a beta version of gggenomes (https://github.com/thackl/gggenomes, copy archived at swh:1:rev:b6b6d8f23fab20a8fd3d9904b37329c914b263a5; *Hackl et al., 2021*).

## Phylogenetic analysis of EMALE and Ngaro proteins

For EMALE core genes, multiple amino acid sequence alignments were constructed with MAFFT using the E-INS-i iterative refinement method (*Nakamura et al., 2018*). Alignments were manually inspected and trimmed to eliminate long insertions and regions of low sequence conservation. The best model and parameters for a maximum-likelihood phylogenetic reconstruction were estimated with modeltest-NG v0.1.6 (*Darriba et al., 2020*) and the tree was computed with IQ-TREE v2.0 (ATPase: LG + I + G4, MCP: LG + R4+ F, Penton: LG + I + G4+ F, Protease: LG + R4+ F, all: -B 1000) (*Minh et al., 2020*). The trees were visualized with ggtree v1.14.6 (*Yu et al., 2017*). For comparison, we also performed Bayesian inference analysis with MrBayes v3.1.2 using the following settings: rates = gamma, aamodelpr = mixed, number of generations = 1 million (*Ronquist and Huelsenbeck, 2003*).

For EMALE tyrosine recombinases, we generated a multiple amino acid sequence alignment with MAFFT v7.310 (`--genafpair`) (*Nakamura et al., 2018*) and an HMM profile (*Eddy, 2011*) from the four sequences present in the type species genomes (EMALE05-EMALE08). We then identified closest relatives with jackhmmer on HmmerWeb v2.40.0 for two iterations (-E 1 `--domE` 1 `--incE` 0.001 `--incdomE` 0.001 `--seqdb` uniprotrefprot) (*Potter et al., 2018*), realigned the EMALE sequences with the 30 best hits (mafft `--genafpair`), and trimmed the alignments with trimAl (-automated1) (*Capella-Gutiérrez et al., 2009*). A maximum-likelihood tree was computed with FastTree v2.1.10 (*Price et al., 2010*). The tree was visualized with ggtree v1.14.6 (*Yu et al., 2017*).

Phylogenies for the Ngaro YRs were generated the same way as for the EMALE YRs, however, only one iteration of jackhmmer was run, and only the top 20 database hits were included in the final tree.

## Comparative analysis of integration sites and their genomic context

For each of the 33 fully resolved EMALEs, we copied up to 10 kb (<10 kb if the EMALE was located within 10 kb of a contig border) of the 5' and 3' host sequences immediately flanking the TIR of the EMALE, and conducted blastn analyses on each of the four *C. burkhardae* genome assemblies with each flanking region separately. In case there was another EMALE located in the 10 kb flanking regions, the second EMALE was omitted from the BLAST search and only host sequence was included. After locating orthologous, and sometimes paralogous, sites in each host strain, we identified TSDs by comparing empty and EMALE-containing alleles using pairwise sequence alignments of homologous sites. To analyze the genomic context of EMALEs for repetitive DNA, flanking regions were analyzed by dot plot analysis.

## Prediction of putative EMALE promoter motifs

We predicted putative promoter motifs in EMALE genomes by running MEME-suite v5.1.1 on all 100 bp upstream regions of all coding sequences. We identified the three highest-scoring motifs for each EMALE type individually. Putative motifs were further validated by analyzing their occurrences

across all EMALE whole genomes, and motifs not consistently present in multiple intergenic regions were excluded.

## Validation of EMALE assemblies by PCR and Sanger sequencing

We designed primer pairs for selected EMALEs to generate overlapping, 700–1100 bp long PCR products. Primer sequences for the validated EMALE01 as shown in *Figure 3—figure supplement 6* are listed in *Supplementary file 1*. PCR products were obtained using 2 ng of genomic DNA template from *C. burkhardae* strain BVI, Cflag, E4-10, or RCC970 in 50 µl total volume containing 5 µl 10 × Q5 Reaction Buffer (NEB, Frankfurt am Main, Germany), 0.5 U of Q5 High-Fidelity DNA Polymerase (NEB), 0.2 mM dNTPs, and 0.5 µM of each primer. Cycling conditions on a TGradient thermocycler (Biometra, Jena, Germany) consisted of: 30 s initial denaturation at 98 °C; 35 cycles of 10 s denaturation at 98 °C, 30 s annealing at 56–60°C (depending on the melting temperature of the respective primers), and 45 s to 1 min extension at 72 °C; followed by a final extension time of 2 min at 72 °C. To check for correct product length and purity, 5 µl of each reaction were mixed with loading dye and analyzed on a 1 % (w/v) agarose gel containing GelRed (VWR, Darmstadt, Germany). The remaining PCR mix was purified using a QIAquick PCR Purification Kit (Qiagen, Hilden, Germany) according to the manufacturer's instructions. Sanger sequencing of PCR products was performed at Eurofins Scientific using the LightRun Tube service. Reads were trimmed, assembled, and mapped to their respective reference EMALEs in Sequencher software v5.2.2 (Gene Codes Corporation, Ann Arbor, MI). Due to the presence of repetitive regions within and across EMALEs in a single host strain and resulting low-quality reads or failed PCRs, the Sanger assemblies typically did not cover the entire length of an EMALE.

## Effect of retrotransposon integration into EMALEs

To test if the insertion of a YR retrotransposon triggered the degeneration of the targeted EMALE, we compared the lengths of conserved genes in Ngaro-containing EMALEs to those without transposon across all 138 EMALEs. Protein lengths were obtained from the sequence files and plotted with a custom R script.

## Acknowledgements

This work was funded by the Max Planck Society and the Marine Microbiology Initiative of the Gordon and Betty Moore Foundation (Grant #5734). We thank I Schlichting and C Roome for support, and A Koslová, J Reinstein, C Bellas, and B Müller for discussions.

## Additional information

### Funding

| Funder | Grant reference number | Author |
|---|---|---|
| Gordon and Betty Moore Foundation | 5734 | Sarah Duponchel<br>Matthias G Fischer |
| Max Planck Institute for Medical Research Heidelberg | | Thomas Hackl<br>Sarah Duponchel<br>Karina Barenhoff<br>Alexa Weinmann<br>Matthias G Fischer |

The funders had no role in study design, data collection and interpretation, or the decision to submit the work for publication.

### Author contributions

Thomas Hackl, Conceptualization, Data curation, Formal analysis, Investigation, Software, Validation, Visualization, Writing – review and editing; Sarah Duponchel, Formal analysis, Visualization, Writing – review and editing; Karina Barenhoff, Investigation; Alexa Weinmann, Investigation, Validation;

Matthias G Fischer, Conceptualization, Formal analysis, Funding acquisition, Investigation, Supervision, Validation, Visualization, Writing – original draft, Writing – review and editing

**Author ORCIDs**
Thomas Hackl http://orcid.org/0000-0002-0022-320X
Matthias G Fischer http://orcid.org/0000-0002-4014-3626

**Decision letter and Author response**
Decision letter https://doi.org/10.7554/72674.sa1
Author response https://doi.org/10.7554/72674.sa2

---

## Additional files

**Supplementary files**
• Supplementary file 1. Sheet 1: Endogenous mavirus-like element (EMALE) statistics. This dataset contains information on each of the 138 EMALEs in four *Cafeteria burkhardae* strains, including their exact location in the host assembly, length, presence of terminal inverted repeats, type score, and Ngaro insertions. Sheet 2: EMALE integration sites. This dataset lists information for each of the 33 fully assembled EMALEs regarding orthologous integration loci in all four host strains, target site duplications, and host genomic context of the integration loci. Sheet 3: Ngaro statistics. This dataset contains information for 80 Ngaro retrotransposons identified in the four *C. burkhardae* assemblies, including their exact location in the host assembly, length, type, and insertion locus (EMALE or eukaryotic chromatin). Sheet 4: Primer sequences. List of oligonucleotides used as PCR and sequencing primers for the validation of EMALE01 RCC970_016B. Numbers in the last column refer to the sequenced PCR products of the assembly diagram shown in *Figure 3—figure supplement 6*, starting with number 1 in the top left corner and ending with number 66 in the bottom right corner.

• Transparent reporting form

### Data availability

Genome sequences of the four *Cafeteria burkhardae* strains have been published previously (PMID 31964893) and are deposited in GenBank (VLTL00000000.1, VLTM00000000.1, VLTN00000000.1, VLTO00000000.1). DNA sequences and genome annotations of endogenous virophages and Ngaro retrotransposons, as well as multiple sequence alignments used for phylogenetic reconstruction are available via (https://github.com/thackl/cb-emales copy archived at https://archive.softwareheritage.org/swh:1:rev:4fa8b383d5f8f207bf2c9d08421fa2239dbbba94). Additional data items are deposited under https://doi.org/10.5281/zenodo.4632783.

The following previously published datasets were used:

| Author(s) | Year | Dataset title | Dataset URL | Database and Identifier |
|---|---|---|---|---|
| Hackl M, Barenhoff D, Heider F | 2020 | Cafeteria roenbergensis strain RCC970-E3, whole genome shotgun sequencing project | https://www.ncbi.nlm.nih.gov/nuccore/VLTL00000000.1 | NCBI GenBank, VLTL00000000.1 |
| Hackl M, Barenhoff D, Heider F | 2020 | Cafeteria roenbergensis strain Cflag, whole genome shotgun sequencing project | https://www.ncbi.nlm.nih.gov/nuccore/VLTM00000000.1 | NCBI GenBank, VLTM00000000.1 |
| Hackl M, Barenhoff D, Heider F | 2020 | Cafeteria roenbergensis strain BVI, whole genome shotgun sequencing project | https://www.ncbi.nlm.nih.gov/nuccore/VLTN00000000.1 | NCBI GenBank, VLTN00000000.1 |
| Hackl M, Barenhoff D, Heider F | 2020 | Cafeteria roenbergensis strain E4-10P, whole genome shotgun sequencing project | https://www.ncbi.nlm.nih.gov/nuccore/VLTO00000000.1 | NCBI GenBank, VLTO00000000.1 |

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
