## [Decision Letter]

**Acceptance summary:**

Approaching the search of novel viruses while in an endogenized stage, rather than as free virions, the study by Hackl et al., reveals a large diversity of complete and fragmented virophage genomes – termed EMALEs – scattered throughout the genomes of four strains of the marine protist *Cafeteria*. Given that the activation of the integrated virophage mavirus during infection by the giant virus, CroV, has been shown to have a protective effect on the *Cafeteria* population, this study provides a tantalizing window into the traces of virophage-giant virus¬-protist interactions in the marine environment. Given the enormous diversity of virophages and giant viruses that have been found in metagenomes with no known hosts, this study is a step towards deciphering the biology of these viruses.

**Decision letter after peer review:**

Thank you for submitting your article "Virophages and retrotransposons colonize the genomes of a heterotrophic flagellate" for consideration by *eLife*. Your article has been reviewed by 3 peer reviewers, and the evaluation has been overseen by a Reviewing Editor and George Perry as the Senior Editor. The following individuals involved in review of your submission have agreed to reveal their identity: Frank Aylward (Reviewer #1); Chantal Abergel (Reviewer #2).

Essential revisions:

The reviewers made several suggestions and comments that, if addressed, would significantly strengthen your manuscript and clarify its presentation, so i recommend that you consider them and revise accordingly. In particular, please address the following comment:

1) All reviewers were curious/raised questions about the evolutionary analyses used to identify EMALE and NGARO endogenization and loss and so some clarification from the authors about the specific points raised in their analytical process would be beneficial to the article.

*Reviewer #1 (Recommendations for the authors):*

Small note on nomenclature. In a few situations the authors use the terms EMALEs and virophage interchangeably, which can be a bit confusing because it is not clear if they are referring to the same thing or if there is some implied difference here (i.e. line 102). This is especially true in the Results section, where it would be clearer to just use EMALE for everything since that term was defined specifically.

For the 7 Ngaro elements in EMALE genes (line 326) it would be useful to have a full breakdown of the predicted function of these genes here. And is this section a discussion of all Ngaro elements or only those found within the 33 high quality EMALEs? That was unclear to me.

It would be useful to note the Pfam domain of the Ngaro elements the first time they are mentioned in the text (or if Pfam is not preferred for some reason some alternative protein family, such as an NCVOG or Interpro ID). This would help to more precisely define these elements for readers. It is also unclear in the Methods how the Ngaro elements were identified- I understand this probably involved manual curation to some extent, but it would be useful to know what features the authors looked for when performing these analyses (best BLASTp hits, Pfam domain hits, etc).

Are homologs of the Ngaro genes found in giant virus genomes? This could be advantageous to giant viruses due to their potential to inactivate virophage. It would be interesting if giant viruses promoted Ngaro gene proliferation. Proteins for a collection of NCLDV and their Pfam annotations are publicly available on the Giant Virus Database (https://faylward.github.io/GVDB/), so it may be easy enough to check what the distribution of these elements across the diversity of NCLDV looks like. If the authors chose to do a quick survey of Ngaro elements in other NCLDV it would increase the breadth and impact of the study, but this is not strictly necessary.

The authors provide an interesting discussion on why a previous study found many fewer endogenous virophage (Blanc et al., PNAS 2015). I reviewed the methods of this study and found them to be quite robust, so it is rather surprising that more endogenous virophage were not found. Hackl et al., rightfully point out that assembly issues may mask the presence of these elements. Another important point is that many fewer reference virophage were available in 2015, so homology based methods were more limited at this time.

The skew towards EMALE integration in intergenic regions is interesting. Given the accumulation of junk DNA in many eukaryotic genomes I would actually predict that many degraded EMALEs would not be removed through purifying selection, and that they would accumulate (though given the high effective population size of these marine hosts it is possible). Is it possible that this bias is due to the difficulty in identifying degraded EMALEs? Perhaps there are more degraded EMALEs in the genome but they are just harder to detect? (see comment below).

Perhaps I missed it but I could not find information on how the EMALEs were initially identified and delineated in the Methods. The results state that "To identify endogenous virophages, we combined sequence similarity searches against known virophage genomes with genomic screening for GC-content anomalies. The two approaches yielded redundant results and virophage elements were clearly discernible from eukaryotic genome regions based on their low (30-50%) GC-content (Figure 1A)". To me this seems quite general and it is not clear what tools were used- for example, I am assuming that sequence similarity searches were done at the amino acid level, and if so was this done with the genes already predicted from the previous Sci Data paper? This is an important detail since gene prediction algorithms often miss NCLDV and virophage genes. In the Scientific Data publication the gene prediction was done with "the BRAKER pipeline which utilizes BLAST Augustus and GeneMark-ES. Augustus and GeneMark-ES gene models were trained with publicly available transcriptomic data of C. roenbergensis E4-10P as extrinsic evidence". If EMALE genes are extensively pseudogenized or not expressed would this pipeline still predict them? More details on EMALE detection and some commentary on the possible limitations of degraded EMALE prediction would be useful.

EVEs necessarily exist on a spectrum from intact to degraded, and the reality is that the most degraded elements may consist of only a single pseudogene and may not be confidently detected using any method. I often think of this, since eukaryotic genomes are full of junk DNA with ambiguous origins- one hypothesis is that much of it derives from endogenous viruses that subsequently degrade beyond all recognition. Some additional discussion of this be useful for interpreting the consequences of retrotransposition, since one would expect that these would actually inactivate quite a few EMALEs and effectively turn them into junk DNA.

*Reviewer #2 (Recommendations for the authors):*

I was only wondering why the authors did not perform an experiment to assess the reactivation of the type 4 EMALEs upon CroV infection as it would nicely demonstrate they are functional MGEs. This would also address the need for another giant virus for EMALEs presenting different promoters.

To me it is not mandatory but would nicely complement the present findings.

The reading of this manuscript also raised one question. Since some groups only possess rve integrase and some present an additional Tyr recombinase, I was wondering if one could be meant for integration into the host genome while the second could be for integration into the giant virus genome, as it is the case for Sputnik which only have the Tyr recombinase and only integrates into the giant virus genome.

I also have a naïve question concerning the pseudogenized EMALEs without Ngaros retrotransposon. The authors addressed this question on the role of Ngaros in EMALE pseudogenization but focused on complete versus absent and did not look for remnants of possible Ngaro. Could pseudogenization in the ones without Ngaros be the result of ancient integrations that disappeared, just leaving for instance the A, B signatures in these EMALEs?

I noticed that in Figure 1 legend B is difficult to read and legends for C and D panels are missing.

In Figure S7, I was wondering what was corresponding to the gap between c023 and c119 for E4-10which appears to be covered by GC content analysis. Same in panel B for RCC970, between c188 and c258 for which a possible explanation is provided, yet the GC content seems to cover this gap.

*Reviewer #3 (Recommendations for the authors):*

This is a very thorough analysis and raises many avenues for future work.

I found myself speculating on how the biology of the system is working in terms of the gain-and-loss of EMALEs, with one idea for an analysis that could be done. I am not proposing this as additional work, but I am curious if the authors had already looked into something along this vein. Presumably complete EMALEs are relatively recent acquisitions while fragmented EMALEs have been inactivated by random processes (point mutations followed by larger insertion-deletions) and are therefore more ancient. Have you looked at making phylogenies for the core morphology module genes in the EMALE fragments to see if they represent more basal types?

Below are some points where some additional details would provide clarification.

– Were there other GC-low regions that did not correspond to EMALEs? It is curious that the genome seems so uniformly GC high.

– Do lone ngaro elements seem to be integrated in GC-low regions?

L410 "either of the three fractions" What are the three fractions? Do the authors mean "host", "EMALE" and "NGARO"?

Why were only PacBio reads used to quantify the percentage of EMALE/NGARO and not also the Illumina data? Was it due to GC bias?

L415 "samtools v1.9(47)" Missing space before the citation.

Figure 1

The size of (B) is a bit small so it is a bit hard to decipher with the symbols.

The description of (B) could do with a better description that states the lines represent contigs in decreasing size ordered from left to right and top to bottom.

The legend is missing descriptions for parts (C and D).

Figure 2

A description in the legend of what the terms in the plot "complete" "FALSE" and "TRUE" refer to is needed.

Figure 3.

Perhaps instead of the dashed line separating the region of homology between types 1 and 2, another line at the 5' end would better show it as a block of homology.

What program was used to produce the genomic maps?

---

## [Author Response]

Essential revisions:The reviewers made several suggestions and comments that, if addressed, would significantly strengthen your manuscript and clarify its presentation, so i recommend that you consider them and revise accordingly. In particular, please address the following comment:1) All reviewers were curious / raised questions about the evolutionary analyses used to identify EMALE and NGARO endogenization and loss and so some clarification from the authors about the specific points raised in their analytical process would be beneficial to the article.

We would like to thank the reviewers and editors for their time and effort to evaluate and improve our manuscript! The comments were very helpful and we have taken great care to address all points accordingly. In particular, we have added two sections to Materials and methods where we describe our procedure of identifying EMALEs and Ngaros in the host genomes (see our responses to reviewer comments 1.1, 1.8). We also extended the discussion towards potential mechanisms of EMALE decay and loss, and modified Figures 1 and Figure 3—figure supplement 3 (previously Figure S7) for more clarity.

Please note we have renamed and re-arranged the supplemental figures to associate each of them with a main figure.

Reviewer #1 (Recommendations for the authors):Small note on nomenclature. In a few situations the authors use the terms EMALEs and virophage interchangeably, which can be a bit confusing because it is not clear if they are referring to the same thing or if there is some implied difference here (i.e. line 102). This is especially true in the Results section, where it would be clearer to just use EMALE for everything since that term was defined specifically.

Thank you for the remark – we tried to eliminate any ambiguous text passages throughout the manuscript.

For the 7 Ngaro elements in EMALE genes (line 326) it would be useful to have a full breakdown of the predicted function of these genes here. And is this section a discussion of all Ngaro elements or only those found within the 33 high quality EMALEs? That was unclear to me.

This section refers to all EMALEs, including partial ones, which is now mentioned in the text. The EMALE genes interrupted by Ngaros are primase/helicase, pPolB, MV12, and 4x MV19, which in addition to Supplementary File 1 (previously Table S3) is now also included in the text. The apparent accumulation in MV19 (predicted peptidase domain) is probably insignificant, since at least two of the affected EMALEs are redundant (found in homologous host genomic loci).

It would be useful to note the Pfam domain of the Ngaro elements the first time they are mentioned in the text (or if Pfam is not preferred for some reason some alternative protein family, such as an NCVOG or Interpro ID). This would help to more precisely define these elements for readers. It is also unclear in the Methods how the Ngaro elements were identified- I understand this probably involved manual curation to some extent, but it would be useful to know what features the authors looked for when performing these analyses (best BLASTp hits, Pfam domain hits, etc).

Pfam domains are now listed in the text. We added a paragraph to Materials and methods called “Detection of Ngaro retrotransposons” that describes our procedure.

Are homologs of the Ngaro genes found in giant virus genomes? This could be advantageous to giant viruses due to their potential to inactivate virophage. It would be interesting if giant viruses promoted Ngaro gene proliferation. Proteins for a collection of NCLDV and their Pfam annotations are publicly available on the Giant Virus Database (https://faylward.github.io/GVDB/), so it may be easy enough to check what the distribution of these elements across the diversity of NCLDV looks like. If the authors chose to do a quick survey of Ngaro elements in other NCLDV it would increase the breadth and impact of the study, but this is not strictly necessary.

We initially had a similar idea, but dismissed it after blast searches against public databases containing giant viruses did not reveal any matches. We now also did a quick comparison of the five Ngaro-type species (1,2a,2b,3,4) elements against the genomes in GVDB. We found one weak hit in one genome (mmseqs translated search, 600 bp, 1.5e-4), but the surrounding regions did not contain any repeat structures that would indicate the presence of a Ngaro-type transposon. Hence, we do not have any evidence that giant viruses contribute to Ngaro proliferation.

The authors provide an interesting discussion on why a previous study found many fewer endogenous virophage (Blanc et al., PNAS 2015). I reviewed the methods of this study and found them to be quite robust, so it is rather surprising that more endogenous virophage were not found. Hackl et al., rightfully point out that assembly issues may mask the presence of these elements. Another important point is that many fewer reference virophage were available in 2015, so homology based methods were more limited at this time.

Indeed, reports of virophage sequences have increased in recent years, which we now point out in the discussion.

The skew towards EMALE integration in intergenic regions is interesting. Given the accumulation of junk DNA in many eukaryotic genomes I would actually predict that many degraded EMALEs would not be removed through purifying selection, and that they would accumulate (though given the high effective population size of these marine hosts it is possible). Is it possible that this bias is due to the difficulty in identifying degraded EMALEs? Perhaps there are more degraded EMALEs in the genome but they are just harder to detect? (see comment below).

You probably meant “The skew towards Ngaro integration in EMALE intergenic regions”? This is an interesting topic, of which we currently know very little about in Cafeteria and thus can only speculate.

First, the genome size of Cafeteria is rather small (~35 Mbp) compared to many other eukaryotes, hence massive accumulation of ‘junk DNA’ is somewhat unlikely.

Second, our study predicts that virophage integration in Cafeteria genomes is an ongoing process. In order to prevent ‘parasitic DNA overload’, mechanisms must exist to either prevent further integration of virophage genomes, or to remove EMALEs from the host genome, e.g. by recombination. This may affect intact as well as degraded elements. It is possible that recombination events leading to EMALE removal increase with increasing numbers of EMALEs, thus restoring a balance faster than via pseudogenization and subsequent loss.

Third, it is not clear how many degraded EMALEs are actually present. We found a few truncated elements (e.g. EMALE type 08), and scattered signs of pseudogenization. However, the few EMALEs with apparent pseudogenes that we were able to PCR-amplify and re-sequence with the Sanger technique turned out to be fully intact, and the premature stop codons were likely an assembly artifact in these cases (Figure 3—figure supplement 6, previously Figure S10). Whereas the actual number of pseudogenes in EMALEs is probably not zero, it is likely to be lower than it may seem at first glance.

Fourth, we did not find evidence that Ngaro insertion leads to pseudogenization in EMALEs.

An alternative possibility is that Ngaros spread via EMALE propagation and that the observed Ngaro integration skew towards EMALE intergenic regions is the result of a selective process that preserves EMALE activity.

Finally, we do not think that we missed degraded EMALEs in our assemblies, please see also our answers to questions 1.1, 1.8 and 1.9.

Perhaps I missed it but I could not find information on how the EMALEs were initially identified and delineated in the Methods. The results state that "To identify endogenous virophages, we combined sequence similarity searches against known virophage genomes with genomic screening for GC-content anomalies. The two approaches yielded redundant results and virophage elements were clearly discernible from eukaryotic genome regions based on their low (30-50%) GC-content (Figure 1A)". To me this seems quite general and it is not clear what tools were used- for example, I am assuming that sequence similarity searches were done at the amino acid level, and if so was this done with the genes already predicted from the previous Sci Data paper? This is an important detail since gene prediction algorithms often miss NCLDV and virophage genes. In the Scientific Data publication the gene prediction was done with "the BRAKER pipeline which utilizes BLAST Augustus and GeneMark-ES. Augustus and GeneMark-ES gene models were trained with publicly available transcriptomic data of C. roenbergensis E4-10P as extrinsic evidence". If EMALE genes are extensively pseudogenized or not expressed would this pipeline still predict them? More details on EMALE detection and some commentary on the possible limitations of degraded EMALE prediction would be useful.

We apologize; this vital method section detailing these aspects should have been included in the initial manuscript. As suspected by the reviewer, EMALEs are not well annotated by eukaryotic annotation pipelines, but have to be annotated separately for good results. To do so, we initially took a manual approach, identified putative endogenous viruses based on clearly visible drops in GC-content along contigs when visualized in Artemis, supported by the presence of terminal inverted repeats in dotplots of those regions. If terminal repeats were missing, we used the GC-content to manually define EMALE boundaries.

For the 33 EMALEs with TIRs, we predicted open reading frames in Artemis and manually curated those into gene calls. For the partial EMALEs we called genes using prodigal with models pretrained on the manually annotated complete EMALEs.

Across the 4 Cafeteria strains, annotated EMALEs account for 85-93% regions with GC-content lower than 56% (measured as 2000bp sequence windows with a GC-content <56%). We manually inspected all locations with deviant GC-content and found the remaining larger regions to be accounted for either by rRNA gene clusters or simple tandem repeats, possibly related to telomeric or centromeric regions.

We also performed blastx searches with the newly identified EMALEs as well as a collection of known virophages against the Cafeteria assemblies to verify that we did not miss other integrated virophages that might not be detectable by GC-content. We are also reasonably confident that our approach is able to detect pseudogenized virophages and that our set of detected EMALEs is comprehensive. However, we of course, cannot exclude that we may have missed viral elements that do not exhibit any obvious deviation in GC-content and at the same time do not share any sequence homology to other known virophages. We modified the discussion to include these considerations.

EVEs necessarily exist on a spectrum from intact to degraded, and the reality is that the most degraded elements may consist of only a single pseudogene and may not be confidently detected using any method. I often think of this, since eukaryotic genomes are full of junk DNA with ambiguous origins- one hypothesis is that much of it derives from endogenous viruses that subsequently degrade beyond all recognition. Some additional discussion of this be useful for interpreting the consequences of retrotransposition, since one would expect that these would actually inactivate quite a few EMALEs and effectively turn them into junk DNA.

In addition to our previous answers, we think that the situation of Cafeteria EMALEs is quite different from other eukaryotic EVEs described so far. Comparison of the four Cafeteria strains revealed that only few EMALE integration loci are conserved among otherwise near-identical host strains. This suggests a fast turnover time for EMALEs, which is not the case for other EVEs that are often described as ‘genomic fossils’. Thus, it may be possible that many EMALEs are removed long before they can pseudogenize, e.g. by TIR-based recombination, homologous recombination between chromosomes, or excised by EMALE- or host-encoded enzymes (integrases, endonucleases). These processes would still apply even when the EMALE is degraded, hence pseudogenized remnants of EMALEs may be rarer than for other eukaryotic EVEs.

Such thoughts are still highly speculative at this early stage of genomic exploration of Cafeteria populations, but we have added a paragraph to the discussion summarizing these considerations.

Reviewer #2 (Recommendations for the authors):I was only wondering why the authors did not perform an experiment to assess the reactivation of the type 4 EMALEs upon CroV infection as it would nicely demonstrate they are functional MGEs. This would also address the need for another giant virus for EMALEs presenting different promoters.To me it is not mandatory but would nicely complement the present findings.

We thank the reviewer for the constructive feedback! We fully agree with this comment and have already invested a considerable amount of work into testing EMALE reactivation in response to CroV infection. However, we strongly prefer to publish these findings separately, as they are quite comprehensive and would clearly exceed the limits of the current manuscript.

The reading of this manuscript also raised one question. Since some groups only possess rve integrase and some present an additional Tyr recombinase, I was wondering if one could be meant for integration into the host genome while the second could be for integration into the giant virus genome, as it is the case for Sputnik which only have the Tyr recombinase and only integrates into the giant virus genome.

Absolutely. This is certainly an interesting notion that we share (see L. 147). Given that virophages need two biological entities (host and giant virus) for successful horizontal propagation, the strategy of physically tracking one of these entities appears to be essential. So far, Sputnik integration has only been shown in mimiviruses, and mavirus integration only in its cellular host. Some EMALEs may have combined both strategies but unfortunately, at this point we lack any data that would shed more light on the biological significance of EMALE-encoded Tyr recombinases.

I also have a naïve question concerning the pseudogenized EMALEs without Ngaros retrotransposon. The authors addressed this question on the role of Ngaros in EMALE pseudogenization but focused on complete versus absent and did not look for remnants of possible Ngaro. Could pseudogenization in the ones without Ngaros be the result of ancient integrations that disappeared, just leaving for instance the A, B signatures in these EMALEs ?

A valid question. We analyzed all EMALE genomes (partial and full) by blast using Ngaro A and B repeats as input and found 5 cases of EMALEs (mostly partial elements) with solo AB repeats. One of them is already marked in Figure 3—figure supplement 1 (previously Figure S2): Cflag_215, which is a truncated EMALE04, and recombination of the Ngaro repeats could have led to the truncation of that element. We now include this information in the text.

However, we found no evidence for increased pseudogenization in these cases.

I noticed that in Figure 1 legend B is difficult to read and legends for C and D panels are missing.

Thank you for noticing! We modified the layout for Figure 1 and added the accidentally deleted legends of C and D. We also rearranged the figure slightly to allow for more space for panel B.

In Figure S7, I was wondering what was corresponding to the gap between c023 and c119 for E4-10which appears to be covered by GC content analysis. Same in panel B for RCC970, between c188 and c258 for which a possible explanation is provided, yet the GC content seems to cover this gap.

You have a keen eye! Contigs c023 and c119 actually overlap, i.e. the ends of those two contigs align to the same region on BVI-028. The appearance of GC/repeat-content covering gaps is a technical artifact from the plotting program. We now manually removed the extra GC-plot regions from Figure 3—figure supplement 3 (previously Figure S7).

Reviewer #3 (Recommendations for the authors):This is a very thorough analysis and raises many avenues for future work.I found myself speculating on how the biology of the system is working in terms of the gain-and-loss of EMALEs, with one idea for an analysis that could be done. I am not proposing this as additional work, but I am curious if the authors had already looked into something along this vein. Presumably complete EMALEs are relatively recent acquisitions while fragmented EMALEs have been inactivated by random processes (point mutations followed by larger insertion-deletions) and are therefore more ancient. Have you looked at making phylogenies for the core morphology module genes in the EMALE fragments to see if they represent more basal types?

We would like to thank the reviewer for the suggestions and the positive assessment of our work!

Yes, we included partial and full EMALEs in our phylogenetic analyses and the resulting patterns were clear: EMALE genes clustered always based on their type, but not based on whether they were found in full or partial EMALEs. That being said, partial EMALEs are not necessarily more degraded (i.e. containing more pseudogenes, insertions or deletions) than full-length EMALEs, but were rather not fully assembled by the computer algorithm (see also answer to question 1.1, reviewer 1). The same pattern was observed for the two truncated EMALEs of type 4 (Cflag_215 and BVI_002), as shown in Figure 4. We did not include the partial EMALEs in these trees for the sake of readability.

Below are some points where some additional details would provide clarification.– Were there other GC-low regions that did not correspond to EMALEs? It is curious that the genome seems so uniformly GC high.

Across the 4 Cafeteria strains, annotated EMALEs account for 85-93% regions with GC-content lower than 56% (measured as 2000bp sequence windows with a GC-content <56%). We manually inspected all locations with deviant GC-content and found the remaining larger regions to be accounted for either by rRNA genes or simple tandem repeats, possibly related to telomeric or centromeric regions. EMALEs, thus, are the main source of low-GC regions in these genomes.

– Do lone ngaro elements seem to be integrated in GC-low regions?

No, the GC-content of regions surrounding Ngaros integrated into the host genome does not differ from the general host GC spectrum. We now state that in the text.

L410 "either of the three fractions" What are the three fractions? Do the authors mean "host", "EMALE" and "NGARO"?

Yes, we now state that explicitly in the text.

Why were only PacBio reads used to quantify the percentage of EMALE/NGARO and not also the Illumina data? Was it due to GC bias?

Exactly, especially the AT-rich EMALEs in the older Illumina runs for E4-10 were only poorly recovered. Later on we used PCR-reduced library preparation protocols which mitigated those biases somewhat, although not entirely. The GC-bias in the Illumina data would misrepresent host-EMALE ratios, low-vs-high-GC EMALE ratios, and due to differences in protocols also ratios between the strains. We added this information to the manuscript.

Figure 1The size of B) is a bit small so it is a bit hard to decipher with the symbols.The description of B) could do with a better description that states the lines represent contigs in decreasing size ordered from left to right and top to bottom.The legend is missing descriptions for parts C) and D).

Thank you for pointing this out! We appended the legend of Figure 1B and added the accidentally deleted legends of C and D. Figure 1B is now a bit larger, since we moved the symbol legend to the lower panel. However, for close examinations it will be inevitable to zoom into the digital version of the figure.

Figure 2A description in the legend of what the terms in the plot "complete" "FALSE" and "TRUE" refer to is needed.

Done.

Figure 3.Perhaps instead of the dashed line separating the region of homology between types 1 and 2, another line at the 5' end would better show it as a block of homology.What program was used to produce the genomic maps?

Thank you for the suggestion! We tried to implement the idea, but were not happy with the result as it made the figure slightly less transparent. Hence, we decided to keep the minimalistic dashed line. We used Adobe Illustrator to create the figure based on imported Artemis screenshots of the annotated EMALEs for scaling.